# ADVERSARIAL EXAMPLES FOR NATURAL LANGUAGE CLASSIFICATION PROBLEMS

## ABSTRACT

Modern machine learning algorithms are often susceptible to adversarial examples — maliciously crafted inputs that are undetectable by humans but that fool the algorithm into producing undesirable behavior. In this work, we show that adversarial examples exist in natural language classification: we formalize the notion of an adversarial example in this setting and describe algorithms that construct such examples. Adversarial perturbations can be crafted for a wide range of tasks — including spam filtering, fake news detection, and sentiment analysis — and affect different models — convolutional and recurrent neural networks as well as linear classifiers to a lesser degree. Constructing an adversarial example involves replacing 10-30% of words in a sentence with synonyms that don't change its meaning. Up to 90% of input examples admit adversarial perturbations; furthermore, these perturbations retain a degree of transferability across models. Our findings demonstrate the existence of vulnerabilities in machine learning systems and hint at limitations in our understanding of classification algorithms.

## 1 INTRODUCTION

Modern machine learning algorithms are often susceptible to adversarial examples — maliciously crafted inputs that are undetectable by humans but that fool the algorithm into producing undesirable behavior. Adversarial examples arise in image classification (Szegedy et al., 2014), speech recognition (Carlini et al., 2016), reinforcement learning (Behzadan & Munir, 2017) and in other domains. The existence of adversarial inputs has obvious security implications and also reveals important shortcomings in our understanding of modern machine learning algorithms.

In this work, we study adversarial perturbations in the context of natural language, and show that common text classification algorithms are affected by adversarially crafted inputs. Our work formalizes the notion of an adversarial example in natural language classification and proposes algorithms for constructing such examples. We also investigate reasons that give rise to adversarial examples, and identify two distinct factors affecting the vulnerability of a model; the first originates in the embedding layer of a neural network classifier, while the second originates in the hidden layers.

Constructing adversarial inputs typically involves replacing 10-30% of words in a sentence with synonyms that don't change its original meaning. Our experiments suggest that such inputs can be consistently constructed across multiple domains — including spam classification, fake news detection, and sentiment analysis — and in different models — convolutional and recurrent neural networks as well as linear classifiers to a lesser degree. Up to 90% of input examples admit adversarial perturbations; furthermore, these perturbations retain a degree of transferability across models.

Our findings suggest the existence of vulnerabilities in text classification systems and hint at limitations in our understanding of these systems. More generally, our work highlights the need to further investigate adversarial inputs in natural language tasks as well as in classification problems over discrete inputs.

---

**Task:** Spam filtering. **Classifier:** LSTM. **Original label:** 100% Spam. **New label:** 89% Non-Spam.

**Text**: your ~~application~~ petition has been ~~accepted~~ recognized thank you for your ~~loan~~ borrower ~~request~~ petition , which we recieved yesterday , your ~~refinance~~ subprime ~~application~~ petition has been ~~accepted~~ recognized good credit or not , we are ready to give you a $ oov loan , after further review , our lenders have established the lowest monthly payments . approval process will take only 1 minute . please visit the confirmation link below and fill-out our short 30 second secure web-form . http : oov

---

**Task:** Sentiment analysis. **Classifier:** CNN. **Original label:** 81% Positive. **New label:** 100% Negative.

**Text**: i ~~went~~ moved to wing wednesday which is all-you-can-eat wings for $ oov even though they raise the prices it 's ~~still~~ ever really great deal . you can eat as many wings you want to get all the different ~~flavors~~ tastes and have a good time enjoying the atmosphere . the girls are smoking hot ! all the types of ~~sauces~~ dressings are awesome ! and i had at least 25 wings in one sitting . i would ~~definitely~~ certainly go again ~~just~~ simply not every ~~wednesday~~ friday maybe once a month .

---

**Task:** Fake news detection. **Classifier:** Naive Bayes. **Original label:** 97% Fake. **New label:** 100% Real

**Text**: trump supporter whose ~~brutal~~ ferocious beating by black ~~mob~~ gangsta was caught on ~~video~~ tape ~~asks~~ demands : " what happened to america ? " [ video ] , " david oov , a 49 year ~~old~~ former ~~chicago~~ rochester man who was brutally beaten by a ~~mob~~ lowlife of black democrats ~~asks~~ demands , " what happened to america ? " here is his very ~~sad~~ disappointing story

---

Figure 1: Adversarial examples for three natural language classification tasks. Replacing a fraction of the words in a document with adversarially-chosen synonyms fools classifiers into predicting an incorrect label. The new document is classified correctly by humans, and preserves most of the original meaning, although it contains small factual and grammatical errors.

## 2 BACKGROUND

We study classification problems, in which the goal is to learn a mapping $f : \mathcal{X} \to \mathcal{Y}$ from an input $x \in \mathcal{X}$ to a target label $y \in \mathcal{Y}$, which lies in some finite set of $K$ classes $\mathcal{Y} = \{y_1, y_2, ..., y_K\}$. The classifier $f$ associates a score $f_{y_k}(x)$ to each class $y_k$ and outputs the class with the highest score. In this paper, $f$ will be parametrized by a deep neural network or a linear model.

### 2.1 ADVERSARIAL EXAMPLES AND IMAGE CLASSIFICATION

Despite recent successes, modern classification algorithms based on deep neural networks are susceptible to adversarial examples (Szegedy et al., 2014), which are maliciously crafted inputs that are indistinguishable from real examples by humans, but that cause the algorithm to misbehave.

In the context of image classification, given a classifier $f$, we say that $x'$ is an adversarial perturbation of $x$ targeting class $y'$ (distinct from the true class $y$ of $x$) if

$$f(x') = y' \text{ and } \|x - x'\| \leq \epsilon. \tag{1}$$

The norm $\|\cdot\|$ captures the notion of an imperceptible perturbation; popular choices include the $\ell_2$ or the $\ell_\infty$ norms. For simplicity, we refer to $x'$ as an adversarial example for $f$.

Adversarial examples can be obtained by solving an optimization problem of the form

$$\max_{x'} J(x') \text{ s.t. } \|x - x'\| \leq \epsilon, \tag{2}$$

in which the objective $J(x')$ measures the extent to which $x'$ is adversarial and may be a function of a target class $y' \neq y$, e.g. $J(x') = f_{y'}(x')$. Algorithms for solving the above objective include the Fast Gradient Sign method or iterative methods based on constrained gradient descent (Goodfellow et al., 2014; Papernot et al., 2016).

### 2.2 CLASSIFYING NATURAL LANGUAGE UTTERANCES

Text classification problems arise in varous domains, including biomedical (Aggarwal & Zhai, 2012), spam filtering (Androutsopoulos et al., 2000), and financial (Schumaker & Chen, 2009).

Linear classifiers with $n$-gram features often perform surprisingly well on text classification benchmarks (Wang & Manning, 2012). In recent years, variants of recurrent networks — especially classifiers based on long short-term memory (Hochreiter & Schmidhuber, 1997) — have helped improve state-of-the-art accuracy; most recently, convolutional neural networks have been shown to be competitive with recurrent methods (Kim, 2014; Zhang et al., 2015)

Natural language classification problems distinguish themselves from image classification by their discrete nature (the inputs $x$ consist of discrete symbols such as characters or words) and by their higher dimensionality, which is typically proportional to vocabulary size. Furthermore, natural language representations are in a sense "higher-level" than image pixels, since they raw words encode significantly more meaning than raw pixel values. These differences pose natural constraints on the notion of an adversarial example, which our work explores.

# 3 ADVERSARIAL EXAMPLES FOR NATURAL LANGUAGE CLASSIFICATION

This work explores adversarial examples in the context of natural language classification. Defining adversarial inputs for text classifiers is complicated by two problems: first, there is no simple notion of metric between utterances (making it difficult to define an imperceptible perturbation); second, discrete inputs are not amenable to gradient-based methods and thus require new optimization algorithms.

In this section, we first propose a general notion of adversarial perturbation that applies to both continuous and discrete inputs. Then, we instantiate this notion in the context of language classification. Finally, we propose a general optimization algorithm for constructing adversarial inputs; the following section examines our approach experimentally.

## 3.1 ALTERED ADVERSARIAL EXAMPLES

A large class of adversarial inputs are formed by adding imperceptible perturbations to ordinary dataset samples. We propose to refer to this general type of adversarial attack as *altered* adversarial examples.

Given a classifier $f$, we say that $x'$ is an adversarial alteration of $x$ targeting class $y'$ if

$$f(x') = y' \quad \text{and} \quad c(x, x') \leq \gamma, \tag{3}$$

for some domain-specific constraint function $c : \mathcal{X} \times \mathcal{X} \to \mathbb{R}_+^L$ and a vector of bounds $\gamma \in \mathbb{R}^L$ that capture the notion of imperceptible alteration via $L \geq 1$ constraints. For example, in the context of chemical molecules, we may use $c$ to capture the edit distance between $x$ and $x'$ or the similarity of the molecules' three-dimensional structure. In the context of image classification, we recover the original notion of adversarial examples by taking $c$ to be an $\ell_2$ or $\ell_\infty$ norm constraint.

Our definition in contrast to other types of adversarial inputs explored in the literature. These include obfuscated examples (Carlini et al., 2016) — in which the input appears as white noise but triggers unwanted behavior (e.g., audio that turns on a smartphone) — and concatenative examples (Jia & Liang, 2017) — in which the input is combined with a distracting sequence that contains irrelevant information. Altered examples, on the other hand, encompass the original notion of adversarial perturbation, and apply in arguably more common settings, such as in classification.

## 3.2 ADVERSARIAL EXAMPLES FOR NATURAL LANGUAGE CLASSIFICATION

In a natural language context, we would intuitively like the altered examples $x'$ to retain the same meaning as the original $x$ .In some settings, it may also be sufficient to generate examples that humans and machines classify into different classes, without requiring that they exactly paraphrase the initial input.

To capture the above intuition, we propose to use a specially-crafted constraint function $c(x, x')$; the goal of this function is to ensure that both utterances share the same meaning and retain common syntactic properties (e.g. the style of writing should remain similar). Specifically, the function $c$ is comprised on two constraints that capture sentence similarity on two levels.

**Semantic similarity.** We propose capturing the meaning of an utterance using the notion of a thought vector (Bengio et al., 2003; Mikolov et al., 2013). A thought vector can be seen as a mapping from sentences to a vector space, in which sentences with similar meanings are close to each other. In this context, our constraint is defined as

$$\|v - v'\|_2 < \gamma_1 \tag{4}$$

where $v$ and $v'$ are thought vectors associated with $x$ and $x'$, respectively, and $\gamma_1$ is a hyper-parameter.

There exist many ways of computing thought vectors for sentences, most of which will be compatible with our optimization algorithm. In the rest of the paper, we will restrict our attention to thought vectors that are averages of the vectors for individual words.

**Syntactic similarity.** Generally, thought vectors do not capture the syntactic validity of a sentence; for example, rearranging all the words in a sentence will produce the same word vector average. To ensure that adversarial sentences are well-formed, we introduce a syntactic constraint, which relies on a language model $P : \mathcal{X} \rightarrow [0, 1]$. Specifically, we require that the language model probability be similar between the perturbed and the original example.

$$|\log P(x') - \log P(x)| < \gamma_2 \tag{5}$$

We suggest training a language model on the same dataset as $f$; this allows the model to capture the extent to which $x$ "looks like" a spam message or a movie review. We also require that the language model probabilities of $x$ and $x'$ be similar: thus if $x$ is an ungrammatical sentence (i.e. a review uses incorrect English), then $x'$ should retain a similar level of correctness.

### 3.3 GREEDY CONSTRUCTION OF ALTERED ADVERSARIAL EXAMPLES

Altered adversarial examples can be obtained by solving an optimization problem of the form

$$\max_{x'} J(x') \ \text{ s.t. } \ c(x, x') \leq \gamma, \tag{6}$$

in which the objective $J(x')$ measures the extent to which $x'$ is adversarial and may be a function of a target class $y' \neq y$, e.g. $J(x') = f_{y'}(x')$. We propose solving this optimization problem approximately using a greedy heuristic outlined as Algorithm 1.

In brief, we propose an iterative procedure that considers at each step all valid one-word changes to a sentence (i.e. which satisfy our constraints) and chooses the one that improves the objective the most. This procedure effectively replaces individual words with their synonyms, resulting in a new sentence of the same length that approximately preserves the original meaning.

---

**Algorithm 1:** Greedy Optimization Strategy for Finding Adversarial Examples

**Data**: Datapoint $x$, termination threshold $\tau$, neighborhood size $N$, parameters $\gamma_1, \gamma_2, \delta$.
We initialize the algorithm at the uncorrupted data point: $x' \leftarrow x$ ;
**while** *Objective is below the threshold $J(x') < \tau$ and fraction of words replaced is less than $\delta$* **do**
    Create a working set $W = \emptyset$ ;
    **for** *each word $w$ in $x$* **do**
        **for** *each word $\bar{w}$ among the $N$ closest to $w$ and different from $w$* **do**
            substitute $w'$ with $\bar{w}$ to get $\bar{x}$ and if $\bar{x}$ satisfies Equ. (5), then $W \leftarrow W \cup \{x'\}$;
    Choose highest scoring world replacement $x' \leftarrow \arg\max_{\bar{x} \in W} J(\bar{x})$ or if $W = \emptyset$, then **break** ;
**return** $x'$;

---

**Algorithm inputs.** Algorithm 1 requires access to a target classifier $f$; it transforms $x$ into $x'$ by optimizing the objective $J$. We assume that $x$ is a set of $n$ discrete symbols called *words* and denoted by $w_i$ for $i = 1, 2, ..., n$. Although we define our algorithm in the context of natural language, it easily extends to general discrete problems as well.

**Optimization strategies.** First, we place a bound $\delta$ on the fraction of words that can be substituted, i.e. $\sum_{i=1}^{n} \mathbb{I}\{w_i \neq w_i'\} \leq \delta \cdot n$; this enables us to "give up" on an example when it clearly does not admit an adversarial alteration. We also set a minimum threshold $\tau$ on the objective (e.g. the minimum desired score of the target label) and terminate when we reach it.

| bad | delicious | enjoy |
|---|---|---|
| inclement | yummy | enjoying |
| mala | scrumptious | enjoys |
| naughty | appetizing | experience |
| rotten | tasty | savor |
| amiss | delectable | savoring |

Table 1: Nearest neighbors in word vector space (Mrkšić et al., 2016).

**Word replacement.** We replace words with their nearest neighbors in a suitable word vector space, and consider the $N$ closest neighbors. Thus, the neighbors are normally words that are likely to occur in the same context as the original word. To ensure that the replacements are also synonyms, we use the GloVE word vectors post-processed by with the method of Mrkšić et al. (2016); this ensures that the vectors satisfy linguistic constraints imposed by known synonym relations, and ensures that words with a similar meaning appear close to each other in the vector space (see Table 1).

## 4 EXPERIMENTS

### 4.1 TASKS

We study adversarial examples on three natural language classification tasks, summarized in Table 2. We held out 10% of the training set for validation; all adversarial examples are generated and evaluated on the test set. Likewise, we trained a trigram language model on the training set of each task and we instantiated the semantic constraint with the word vectors of Mrkšić et al. (2016). We describe our three classification tasks below.

| Dataset | Task | #Train | #Test |
|---|---|---|---|
| Trec07p | Spam filtering | 67.9k | 7.5k |
| Yelp | Sentiment analysis | 560k | 38k |
| News | Fake news detection | 5.3k | 1.0k |

Table 2: Summary of datasets and tasks

**Spam filtering.** The TREC 2007 Public Spam Corpus (*Trec07p*) contains 50,199 spam emails and 25,220 ham (non-spam) emails. We preprocess the data by removing all meta data and HTML tags. There is no standard split for this dataset, so we randomly pick 10% as a test set.

**Sentiment analysis.** The Yelp Review Polarity dataset (*Yelp*; Zhang et al., 2015) consists of almost 600,000 customer reviews from Yelp, covering primarily restaurant reviews. Each review is labeled as either positive or negative.

**Fake news detection.** The News dataset (McIntire, 2017) contains 6,336 articles scraped from online sources, and includes both real and fake news. Each article contains a headline and body text (which we concatenated before classification) and is associated with a binary label.

### 4.2 MODELS

We study adversarial example on a range of models that are widely used for classification; these include both linear classifiers and state-of-the-art deep learning algorithms.

**Naive Bayes.** This linear model has a long history in text classification and it is still popular for its simplicity. We convert each document into a bag-of-words representation, and following Wang & Manning (2012), we binarize the word features and use a multinomial model for classification.

**Long short-term memory.** Long-short term memory (LSTM; Hochreiter & Schmidhuber, 1997) is widely used in sequence modeling. We built a single-layer LSTM with 512 hidden units as in Zhang et al. (2015). The input to the LSTM is first transformed to a 300-dimensional vector using pretrained `word2vec` embeddings (Mikolov et al., 2013). We then average the outputs of the LSTM at each timestep to obtain a feature vector for a final logistic regression to predict the sentiment.

**Shallow word-level convolutional networks.** An alternative approach to text classification are convolutional neural networks (CNNs; Kim, 2014) We train a CNN with an embedding layer (as in the LSTM) a temporal convolutional layer, followed by max-pooling over time, and a fully connected layer for classification. We use a uniform filter size of 3 in each convolutional feature map; all other settings are identical to those of Kim (2014).

**Deep character-level convolutional networks.** We implement the character-level network of Conneau et al. (2016), which includes 4 stages. Each stage has 2 convolutional layers with batch normalization and 1 max-pooling layer; convolutional and pooling layers have strides of 1 and 2, respectively and filters of size 3. We start with 64 feature maps, and double the amount after each pooling step, concluding with k-max pooling layer with $k = 8$. The resulting activations in $\mathbb{R}^{4096}$ are classified by 3 fully connected layers.

### 4.3 MAIN EXPERIMENTS

Table 3 shows the accuracy of each classification model on the three datasets as well as on adversarial inputs generated using Algorithm 1. We manually selected the optimization settings that led to a reasonable tradeoff between the strength and the coherence of the adversarial examples. Specifically, in all experiments, we used a target of $\tau = 0.7$, a neighborhood size of $N = 15$, and parameters $\gamma_1 = 0.2$ and $\delta = 0.5$; we set the syntactic bound to $\gamma_2 = 2$ nats for sentiment analysis and fake news detection and $\gamma_2 = \infty$ for spam; spam messages were often malformed an the language model was no longer useful. We also compare against random perturbations obtained by replacing the $\arg\max$ in Algorithm 1 with random sampling.

| Data | | NB | LSTM | WCNN | VDCNN |
|---|---|---|---|---|---|
| | CLN | 97.1% | 99.1% | 99.7% | |
| Trec07p | RND | 97.7% | 98.6% | 99.6% | |
| | ADV | 15.1% | 39.8% | 64.5% | |
| | CLN | 87.9% | 95.3% | 94.9% | 95.1% |
| Yelp | RND | 86.8% | 94.5% | 94.7% | 93.1% |
| | ADV | 9.0% | 24.0% | 39.0% | 53.0% |
| | CLN | 91.0% | 93.0% | 96.0% | 93.4% |
| News | RND | 84.0% | 94.6% | 93.3% | 92.7% |
| | ADV | 9.0% | 37.0% | 71.0% | 11.0% |

Table 3: Classifier accuracy on each dataset. CLN, RND, and ADV stand for clean, randomly corrupted, and adversarially corrupted inputs, respectively.

All models are susceptible to adversarial examples to a certain degree, which depends in part on the task. Certain problems, such as spam filtering seem easier to classify and are less amenable to adversarial inputs; conversely, it is easier to fool the models on more difficult tasks, such as fake news detection. All methods are equally robust to random perturbations (just as image classifiers typically are), suggesting that adversarial inputs reside in very specific directions off the manifold of normal samples.

### 4.4 HUMAN EVALUATION

| Input | Trec07p | Yelp | News |
|---|---|---|---|
| Original | 87% | 93% | 64% |
| Adversarial | 93% | 87% | 58% |

Table 4: Human classification accuracy on adversarial examples for the LSTM model.

We verified the quality and the coherence of our examples via human experiments on Amazon Mechanical Turk. First, we subsampled a 100 random test set examples and asked human evaluators to assign labels (e.g. positive or negative review) to both the original data points, and their adversarially altered versions. We averaged the opinions of five different evaluations for each query. We found that human evaluators achieved similar accuracies on both types of inputs, suggesting that our adversarial alterations preserved key semantics sufficiently well to be recognized by a human. Human accuracy generally falls below that of the algorithms: the fake news task is inherently difficult, while non-spam email is often misclassified since there is no standard definition for "ham" emails; on sentiment analysis, both accuracies are within a reasonable margin of error.

Next, we asked human annotators to rate the "writing quality" of the same set of examples on a scale of one to five, with five being the highest possible quality and likely generated by a human,

and one being the lowest quality, likely generated by a machine. Table 5 shows that humans tend to assign similar scores to both sets of samples. Although our adversarial examples were not perfectly formed, these results suggest that they were of comparable quality to the original examples (which also contained multiple spelling and grammar errors).

## 4.5 ERROR ANALYSIS

We found that our adversarial examples exhibit three kinds of errors: syntactic, semantic, and factual. Syntactic errors are ungrammatical word substitutions; these include replacing "isis claim responsibility for shooting" with "isis petition responsibility for shooting" and "never before has an fbi director" to "never until has an fbi director"; the first error is due to multiple word meanings, while the latter is due to the words being unrelated (and far in word vector space).

| Input | Trec07p | Yelp | News |
|---|---|---|---|
| Original | 2.64 | 2.37 | 2.72 |
| Adversarial | 2.75 | 2.38 | 2.47 |

Table 5: Human classification accuracy on adversarial examples for the LSTM model.

Semantic errors arise when the meaning of a sentence is altered. Most often, this is due to multiple word senses — e.g., "isis claim responsibility for shooting" to "isis claim responsibility for filming" — or due to word embedding errors — e.g., "isis claim responsibility for ceasefire". Factual errors are a special case when the sentence becomes obviously false, e.g. when chainging "Monday, March 16" changed to "Thursday, March 16", or "FBI assistant director james kallstrom" to "Pentagon assistant director james kallstrom", or "republicans backing Trump" to "republicans backing Obama". These may not be an issue with fake reviews or fake news, and may be remedied via specialized techniques, e.g. by performing Named Entity Recognition.

## 4.6 TRANSFERABILITY

|  | NB | LSTM | WCNN | VDCNN |
|---|---|---|---|---|
| NB | 20% | 77% | 75% | 88% |
| LSTM | 67% | 17% | 64% | 83% |
| WCNN | 63% | 64% | 17% | 84% |
| VDCNN | 77% | 85% | 87% | 23% |

Table 6: Transferability of adversarial examples on the Yelp dataset. Row $i$ and column $j$ show the accuracy of adversarial samples generated for model $i$ evaluated on model $j$.

An intriguing property of image classification models is that adversarial examples generated for one classifier are likely to be misclassified by ther classifiers. We also examined whether adversarial texts transfer between the four models, focusing on the Yelp dataset. As seen in Table 6, there is a moderate degree of transferability between models. Interestingly, adversarial examples for three word level models (NB, LSTM, WCNN) do not generalize as well to the character level deep CNN as to other word level models, which suggests that the choice of input representation (character or word) is a factor that affects transferability.

## 4.7 EXPLAINING ADVERSARIAL EXAMPLES

We attribute the existence of adversarial examples to two factors, which we refer to as embedding-based and representation-based. A neural network classifier for text contains two stages: first, an embedding layer maps discrete words into continuous vectors; then, the embeddings are classified via convolutional or fully-connected layers.

Representation-based errors arise in the higher layers of the network; these layers are very similar to those of image classifiers and are therefore susceptible to the same types of attacks. In other words, by replacing a word, we may adversarially push the embedding representation off the learned manifold into an adversarial region, causing a mislabel. Representation-level adversarial perturbations have been studied by Miyato et al. (2017) in the context of adversarial training, and we observed them in our experiments as well.

On the other hand, embedding-based errors can be attributed to the inherently high dimensionality of the vocabulary, and can be studied by looking at linear classifiers. For example, a perturbation for Naive Bayes implies that words which we consider equivalent (e.g., definitely and certainly)

occur with different frequencies among the two classes. This, in turn, can happen for two reasons: firstly, since the vocabulary is large, such words will arise due to statistical noise, unless the dataset is also very large; secondly, two words may truly have different conditional class probabilities, either because of multiple meanings (e.g., awfully great deal and awfully bad) or because of hidden patterns that we don't understand (dogs are truly mentioned more often in spam than cats).

# 5 DISCUSSION

## 5.1 APPLICATIONS OF LANGUAGE-BASED ADVERSARIAL EXAMPLES

Our work demonstrates the existence of adversarial examples in state-of-the-art models for spam and sentiment classification (the fake news state-of-the-art is not yet established). The existence and transferability of such examples (obtained with very simple methods), hint at the existence of vulnerabilities in a number of systems; thes include text filtering systems (e.g., spam, racism), online ranking algorithms, speech command processors, and others. More generally, together with the work of Jia & Liang (2017), our observations lend further evidence to the prevalence of adversarial attacks in the natural language domain.

On the other hand, adversarial inputs can also improve algorithms via adversarial training (Shrivastava et al., 2016) by serving as extra data and thus increasing performance and robustness to adversarial attacks. Miyato et al. (2017) showed that adversarial perturbations to word embeddings are useful for semi-supervised learning; our findings hint at the possibility of adversarial training in the space of words.

## 5.2 COMPARISON TO OTHER DOMAINS

We found that adversarial examples are somewhat less abundant than in computer vision, especially on "easy" tasks such as spam classification. Interestingly, they affect linear models more; this smaller gap can be explained by the fact that text classification models are relatively shallow and word inputs are in a sense "higher-level" than pixels: they are more susceptible to perturbations and leave less work to the rest of the network. Although defining a metric between utterance is non-trivial, it is also somewhat more forgiving: in many domains (such as spam) grammatical errors are common, and the meaning does not get lost (or seem unnatural) with errors.

Finally, it's interesting to note that language-based problems have more direct access to the system being attacked; adversarial images are typically processed by real-world sensors, which affects their strength Kurakin et al. (2016); language-based examples are fed into the system directly, which could make them more potent.

## 5.3 FUTURE WORK

Our results demonstrate the existence of natural-language adversarial perturbations. However, our simple perturbations could be improved via a more sophisticated algorithm that takes advantage of language processing technologies, such as syntactic parsing, named entity recognition, or paraphrasing, or that could be assisted by humans. Furthermore, the existing search procedure naturally generalizes to beam search, and could modify phrases rather than words. Interesting extensions apply to character-based substitutions, targeting both word- and character-based systems.

# 6 CONCLUSION

We generalize the concept of adversarial examples to natural language classification by proposing a simple yet effective similarity metric for text. Then evaluate our approach on several classification tasks and show that a simple greedy algorithm is effective at finding adversarial examples in each setting.

The presence of adversarial examples for text classification poses threat to real-world machine learning systems. We further study of adversarial examples for text classification with help defend these systems and improve the accuracy of classification algorithms via adversarial training.

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

---

**Classifier:** LSTM. **Original label:** 91% Fake News. **New label:** 1% Fake News.

**Text:** ~~difference~~ discrepancy between growing up in the 1960s compared to 2016 , " you are here : home / us / ~~difference~~ discrepancy between growing up in the 1960s compared to 2016 difference between growing up in the 1960s compared to 2016 october 27 , 2016 pinterest seth oov reports that in august of this year , campus carry ~~went~~ moved into effect on texas ' ~~public~~ demographic college campuses , ~~enabling~~ authorizing students and staff ~~with~~ among valid concealed handgun licenses to legally carry their firearms . predictably , ~~leftists~~ democrats freaked out at the idea of people legally carrying firearms in their " safe spaces . " as we reported back in august , the most famous form of protest on texas college campuses was " oov not oov , " a movement where students who ~~opposed~~ objected campus carry ~~took~~ picked adult sex toys with them all across the campus . related : campus carry starts ~~today~~ monday in texas ; here 's how liberal students are protesting ... despite these oov , campus carry is in effect in texas , and there is not mass murder happening in oov , classrooms , or professors ' offices . who 'd have oov , right ? well ,

---

**Classifier:** Naive Bayes. **Original label:** 96% Fake News. **New label:** 0% Fake News.

**Text:** israel votes : netanyahu 's last-ditch vow to his ~~base~~ foundation - a dead peace process ( +video ) , " politicians make many ~~campaign~~ movements promises they do n't intend to ~~deliver~~ render on . but netanyahu 's promise ~~monday~~ thu to never ~~agree~~ subscribe to a palestinian state fits his record . israeli prime minister benjamin netanyahu talks as he visits a construction site in oov oov , east jerusalem , ~~monday~~ thu march 16 , 2015 , a day ahead of legislative elections . netanyahu is seeking his ~~fourth~~ iii term as prime minister . ~~with~~ via israel 's final pre-election polls ~~pointing~~ portraying to a difficult road for prime minister benjamin netanyahu to stay in power , he spent his final days on the campaign trail throwing red meat to his ~~base~~ foundation . oov oov warned israeli voters that only mr. netanyahu has the strength to stand up to " " hussein obama . " " ~~campaign~~ movements ~~ads~~ advertisement compared israeli oov workers and regulators to hamas militants and called his opponents tools of shadowy foreign financiers ( a strange charge given his own close ties to us ~~casino~~ poker billionaire sheldon adelson ) . but on monday the prime minister delivered his show oov : vote

---

**Classifier:** WordCNN. **Original label:** 91% Fake News. **New label:** 1% Fake News.

**Task:** " we must smash the clinton machine : democratic elites and the media sold out to hillary this time , but change is coming " , " a times story headlined " ~~obama~~ gingrich ~~privately~~ stealthily ~~tells~~ narrates ~~donors~~ contributors time is coming to unite behind hillary " had ~~obama~~ gingrich telling dnc high oov to " come together . " in it obama " did n't explicitly call on sanders to quit " but a " white house official " confirmed his " unusually candid " words . it was a plant dressed up as a scoop . obama spoke not privately but on background , and not to his ~~donors~~ contributors but ~~through~~ via them ( and the paper ) to his base . it was a different portrait of obama as oov : political , financial and media ~~elites~~ oligarchs , all working as one to put down a revolt . ~~obama~~ gingrich 's neutrality is a polite scam . his " private " chat ~~came~~ entered before voters in 29 states even had their say . presidents never let appointees make endorsements , but three obama cabinet secretaries – ~~agriculture~~ husbandry 's tom vilsack , oov 's julian castro and labor 's thomas perez – backed clinton

Figure 2: Examples of adversarial text generated for Fake News Detection

## A    APPENDIX

We include multiple examples of adversarially perturbed inputs in this appendix. We provide examples for each model and each task.

**Classifier:** Naive Bayes. **Original label:** 90% Negative. **New label:** 18% Negative.

**Text:** i ordered a carne asada burrito and it was ~~garbage~~ junk ! the carne asada tasted bad , thin and hard , just bad quality . ~~roberto~~ enrico 's is not that great but it 's better than this place

---

**Classifier:** LSTM. **Original label:** 97% Negative. **New label:** 0% Negative.

**Text:** ~~this~~ that ~~place~~ location is far from the the best pho experience i 've ever had ( that is almost a bad pun ) . it 's really not bad , but there are much better vietnamese restaurants in vegas . the pho broth is n't on the same level as pho so 1 or lemongrass cafe . for some reason , they were out of bean ~~sprouts~~ sprout and while i do n't love them , i 've become accustomed to having them in my pho . finally , i was a little disappointed that they do n't serve tripe in any of their pho variations . overall , although i did enjoy the soup , i probably wo n't return . i need to try the ~~jenni~~ jenny pho place just down the street . if that does n't work out , i 'll just have to make the extra drive to chinatown .

---

**Classifier:** WordCNN. **Original label:** 90% Negative. **New label:** 10% Negative.

**Text:** give this ~~place~~ location 2 stars because of the new car buying experience . i have always ~~owned~~ possessed an acura and never really encountered problems with buying a new car . i was referred to a salesman chip , who by the way is amazing ! ! ! he was n't there at the time and i wanted to purchase a brand new car , so i saw the next salesman named ~~steve~~ craig w. he was just as amazing as chip . he worked with us throughout our process and was incredibly patient with us . at the end of our car buying , we had a bump in the road and had a huge misunderstanding about the price of the car . steve w. ~~remained~~ retained professional and patient with us . when the sales manager , ~~chris~~ kyle b. came out , he was n't happy with us . he was a tad bit short tempered with us and made us feel like crap . it ruined my experience with buying a car with acura , which is probably the reason why i am looking in henderson or california to buy ( has a much more competitive price as well

---

**Classifier:** VDCNN. **Original label:** 94% Negative. **New label:** 8% Negative.

**Text:** i ~~wanted~~ want to love you spicy pickle , but it was your ~~pickle~~ dill i ~~liked~~ enjoyed best . i had the bandito ~~panini~~ gorgonzola with peppered turkey , pepperjack , roasted red peppers , sundried tomatoes , chipotle mayo . i took the sandwich to go and by the time i got to eat , it was pretty soggy . the blend of flavors did n't save it . i 'll have to give it a second chance and dine in , maybe try a specialty sandwich ~~instead~~ equally .

---

**Classifier:** LSTM. **Original label:** 97% Negative. **New label:** 0% Negative.

**Text:** long lines but ~~amazing~~ surprising ~~burger~~ cheeseburger and ~~fries~~ chips as ~~always~~ consistently . i always get the double double and the ~~fries~~ nuggets animal style . you can not go wrong with that . manager was super ~~sweet~~ sugary and ~~nice~~ good

---

**Classifier:** Naive Bayes. **Original label:** 99% Negative. **New label:** 0% Negative.

**Text:** must preface this review by saying that this is the only time i visited nicky 's thai kitchen ( or pittsburgh for that matter ) , so it could just be a fluke - maybe the chef had a bad day - but this is the second ~~worst~~ largest thai food i 've had anywhere in the world ... the ~~worst~~ largest was in aruba . i 'm a fan of thai food as you can tell - craving it even when on vacation in aruba - so i have a certain expectation when i saw the 4-star ratings for this restaurant . but was n't i ~~disappointed~~ disappoint ! we ordered drunken noodles and panang curry , both with beef , and spicy . typical thai dish right ? wrong . what we got are two ~~tasteless~~ dorky dishes . the drunken noodles dish is not just bland and way too mild , but both the veggie and meat tasted ~~stale~~ old . the panang curry was equally ~~unimpressive~~ bland . the color of the broth may be right , but there is only a hint of curry taste in it . the meat was ~~chewy~~ succulent to the point that i gave up on

Figure 3: Examples of adversarial text generated for Sentiment Analysis

**Classifier:** Naive Bayes. **Original label:** 99% Spam. **New label:** 0% Spam.

**Text:** wondercum is a wonderful combination of fine ~~herbs~~ weed extract that are well known for centuries we do not have any branched or ~~stores~~ storing located ~~anywhere~~ whenever . http : oov

---

**Classifier:** LSTM. **Original label:** 73% Spam. **New label:** 0% Spam.

**Text:** ~~view~~ viewpoint ~~pics~~ images of ~~christian~~ protestant singles in your ~~area~~ realm ~~meet~~ cater ~~christian~~ protestant singles with oov values in your ~~area~~ realm . oov this ~~email~~ mailroom is a commercial ~~advertisement~~ publicity ~~sent~~ forwarded in compliance with the oov act of 2003. we have no ~~desire~~ volition to send you information that is not wanted , ~~therefore~~ similarly , if you wish to be excluded from future mailings , please use the link at the bottom of the page

---

**Task:** Spam Classification. **Classifier:** WordCNN. **Original label:** 89% Spam. **New label:** 0% Spam.

**Text:** your ~~loan~~ borrower ~~application~~ apps is ~~waiting~~ hoping ~~dear~~ pricey ~~homeowner~~ landowner are you still paying too much for your current ~~mortgage~~ subprime ? refinaance us best ~~rate~~ cadence . your ~~approval~~ ratification is ~~waiting~~ expecting . please ~~respond~~ cater oov ~~http~~ myspace : oov ~~helen~~ edith gay lendingtree ~~department~~ administration

---

**Task:** Spam Classification. **Classifier:** NB. **Original label:** 98% Spam. **New label:** 0% Spam.

**Text:** urgent : your paypal account has expired ! paypal body , td protect your account ¡oov¿ sure you never provide your password to ~~fraudulent~~ bogus websites . for more information on protecting yourself from fraud , please review our security tips at https : ¡oov¿ your ¡oov¿ should never give your paypal password to anyone , including paypal ~~employees~~ gov . upgrade your information dear ~~member~~ lawmakers , it has come to our attention that your paypal ~~billing~~ legislation information is out of date . therefore we have had to put a limit your paypal account . we require you to update your ~~billing~~ legislation information on or before 4th june 2007. failure to update your records may result in a suspension of your account . to update your paypal ~~billing~~ invoices information click the link below , login to your account with your email address and password and read the on screen instructions : http : //www.paypal.com/cgi-bin/webscr ? ¡oov¿ this security measure helps us continue to offer paypal as a secure and cost-effective payment service . we appreciate your cooperation and assistance . sincerely , the paypal team please do not reply to this email . this mailbox is not ~~monitored~~ oversight and you will not

---

**Task:** Spam Classification. **Classifier:** WordCNN. **Original label:** 98% Spam. **New label:** 68% Spam.

**Text:** this job offer is just for you ! ~~dear~~ pricey ~~sirs~~ gentlemen , ~~aegis~~ sponsorship capital group llc ( aegis ) is a ~~specialty~~ expert ~~investment~~ capital firm managing private ~~equity~~ fairness and ~~venture~~ enterprise capital funds ~~with~~ into a ~~national~~ nationalist focus on small businesses and the social ~~benefits~~ advantages of supporting ~~entrepreneurs~~ corporations and ~~enhancing~~ reinforcing local job ~~creation~~ introduction . we ~~would~~ should like to stress , that our ~~company~~ enterprise pays ~~special~~ peculiar ~~attention~~ concentration to customer support of ~~private~~ particular ~~customers~~ subscribers , though we also have the corresponding business plans for the bigger companies as ~~well~~ correctly . a ~~more~~ wider detailed information about our ~~company~~ enterprise you may obtain at our ~~official~~ formal ~~website~~ venue . due to the necessity for ~~expansion~~ enlargement of our ~~company~~ enterprise , we have announced some additional openings for new ~~employees~~ officials . we are ~~glad~~ contented to ~~offer~~ supply you one of the vacant positions in our ~~company~~ business team a ~~position~~ stance of the ; ~~account~~ accountant ~~manager~~ admin .you will have the responsibility for the following ~~duties~~ obligations : ~~fulfillment~~ implementation of ~~orders~~ commandments given by the ~~company~~ enterprise , operations with the ~~bank~~ banco ~~transfers~~ assignments ( direct ~~deposits~~ filings and ~~wires~~ threads ) ~~from~~ into ~~customers~~ subscribers , implementation of ~~calculations~~ computations

Figure 4: Examples of adversarial text generated for Spam Classification

