# OpenReview forum: "Adversarial Examples for Natural Language Classification Problems"
_ICLR.cc/2018/Conference — Reject_

### Official Review · AnonReviewer3 · 2017-11-15
**Nice overview of adversarial techniques in natural language classification**

**Rating:** 6
**Confidence:** 3

**Review:**

Nice overview of adversarial techniques in natural language classification. The paper introduces the problem of adversarial perturbations, how they are constructed and demonstrate what effect they can have on a machine learning models.

The authors study several real-world adversarial examples, such as spam filtering, sentiment analysis and fake news and use these examples to test several popular classification models in context of adversarial perturbations.

Their results demonstrate the existence of adversarial perturbations in NLP and show that several different types of errors occur (syntactic, semantic, and factual). Studying each of these errors type can help defend and improve the classification algorithms via adversarial training.

Pros: Good analysis on real-world examples
Cons: I was expecting more actual solutions in addition to analysis

---

> ### Author Response · Authors · 2017-12-16
> **Could you please elaborate on why our score is low?**
>
> Thank you for your generally positive review. However, since the score that you are giving us (6.0) is not very high, could you please elaborate on your concerns with this paper?
>
> We would be very happy to improve the paper before the final decision period, but your only negative comment is that you were expecting "more actual solutions in addition to analysis". We don’t know how to interpret that.
>
> If you are interested in ways of protecting against attacks, we tried using the method of Miyato et al., which results in a small increases in performance on adversarial examples.

---

> ### Author Response · Authors · 2017-12-30
> **Have you had the chance to read our response?**
>
> Happy holidays! Since the end of the discussion period is approaching, we wanted to check if you had the chance to read our response. We would like very much to address your concerns and have a discussion about our paper before the January 5 deadline.
>
> Thank you!

---

### Official Review · AnonReviewer1 · 2017-11-28
**No comparison to existing work**

**Rating:** 4
**Confidence:** 5

**Review:**

This paper proposes a method to generate adversarial examples for text classification problems. They do this by iteratively replacing words in a sentence with words that are close in its embedding space and which cause a change in the predicted class of the text. To preserve correct grammar, they only change words that don't significantly change the probability of the sentence under a language model.

The approach seems incremental and very similar to existing work such as Papernot et. al. The paper also states in the discussion in section 5.1 that they generate adversarial examples in state-of-the-art models, however, they ignore some state of the art models entirely such as Miyato et. al.

The experiments are solely missing comparisons to existing text adversarial generation approaches such as Papernot et. al and a comparison to adversarial training for text classification in Miyato et. al which might already mitigate this attack. The experimental section also fails to describe what kind of language model is used, (what kind of trigram LM is used? A traditional (non-neural) LM? Does it use backoff?).

Finally, algorithm 1 does not seem to enforce the semantic constraints in Eq. 4 despite it being mentioned in the text. This can be seen in section 4.5 where the algorithm is described as choosing words that were far in word vector space. The last sentence in section 6 is also unfounded.


Nicolas Papernot, Patrick McDaniel, Ian Goodfellow, Somesh Jha, Z.Berkay Celik, and Ananthram Swami
Practical Black-Box Attacks against Machine Learning.
Proceedings of the 2017 ACM Asia Conference on Computer and Communications Security

Takeru Miyato, Andrew M. Dai and Ian Goodfellow
Adversarial Training Methods for Semi-Supervised Text Classification.
International Conference on Learning Representation (ICLR), 2017

* I increased the score in response to the additional experiments done with Miyato et. al. However, the lack of a more extensive comparison with Papernot et. al. is still needed. The venue for that paper might not be well known but it was submitted to arXiv computer science too and the paper seems very related to this work. It's hard to say if Papernot et. al produces more perceptible samples without doing a proper comparison. I find the lack of a quantitative comparison to any existing adversarial example technique problematic.

---

> ### Author Response · Authors · 2017-12-16
> **Please provide references to justify your claim of "no comparison to previous work"**
>
> Thank you very much for your feedback. We see that you raise several issues in your review.
>
> 1. We are not evaluating our method on state-of-the-art models
>
> Here, we respectfully disagree: this claim is incorrect.
>
> Our models (except the linear classifier) achieve accuracies of 94.9%-95.3% on the popular Yelp dataset (same as the papers whose models we used). The 2017 state-of-the art is ~97% using a ResNet [1], and the 2016 SOA was 96%. On the widely used IMDB dataset we obtain accuracies of ~92-93%; the state-of-the-art around 96%. (We didn't include IMBD results due to lack of space and similarity to Yelp). On spam detection, we also obtain nearly perfect accuracy. Finally, there is no standard fake news dataset, but we achieve high accuracy on the one that we use.
>
> Furthermore, our architectures are very modern and date from as recently as last year (see our citations)
>
> [1] Johnson and Zhang, http://www.aclweb.org/anthology/P17-1052
>
> 2. We do not take into account the recent work of Miyato et al.
>
> First, note Miyato et al. propose a method for adversarial training, which is very different from adversarial examples (what we study). Adversarial training is at the moment not part of the standard toolkit for classification algorithm, which is why we did not immediately compare to it.
>
> You also mention that the method of Miyato et al.. could be used as a defense. However, that too is not correct: our adversarial examples arise from large perturbations in embedding space (we replace an entire word); Miyato et al., on the other hand, perform adversarial training in embedding space, which consists in introducing very small perturbations. In the context of Naive Bayes, their method does not even apply (there are no word embeddings in NB)
>
> We confirmed this empirically by testing the method of Miyato et al. on the CNN model. We observed only a small (10%) improvement in accuracy on AEs. We are also currently running additional experiments on every setup. We will report here our final results once they are done.
>
>
> 3. There is no comparison to existing adversarial text generation approaches
>
> We are more than happy to compare to any existing work. However, the Papernot et al. paper you cite has no mention of text classification at all. The underlying algorithm is gradient-based, and is not applicable to discrete inputs (relative to which we cannot differentiate the model). The Miyato paper you mention does not work well as a defense against our method (see above).
>
> An anonymous commenter mentioned some relevant work; please see also our detailed response to their comment.
>
> 4. Extra technical questions and clarifications
>
> The language model we use a tri-gram model. This is a detail that we forgot to mention and that we will add into the paper.
>
> We certainly enforce equation (4) in our algorithm. There is a typo in Algorithm 1 (it should read "Equations 4, 5" instead of "Equation 5"), which we will correct right away. We apologize for any confusion due to this typo.

---

> > ### Comment · AnonReviewer1 · 2017-12-16
> > **Correction to reference**
> >
> > Apologies, from your comment I realised I cited the wrong Papernot et. al paper for previous work on adversarial text generation. I meant to cite:
> >
> > Papernot, N., McDaniel, P., Swami, A., & Harang, R. (2016, November). Crafting adversarial input sequences for recurrent neural networks. In Military Communications Conference, MILCOM 2016-2016 IEEE (pp. 49-54). IEEE.

---

> > > ### Author Response · Authors · 2017-12-16
> > > **Comparison to the new reference**
> > >
> > > Thank you for pointing us to this paper. We were not aware of it, and we agree that it needs to be cited.
> > >
> > > However, the scope of this paper is very limited compared to our work.
> > >
> > > 1. Most importantly, there is no notion of preserving the meaning of the original sentence.
> > >
> > > In other words, Papernot et al. replace words in a sentence without ever looking at whether the new words are related to the originals. In our experience, this would most likely produce non-sensical sentences, and a human would recognize them as such. The paper also offers no evaluation of the constructed AEs besides the accuracy of the classifier, e.g. it doesn't evaluate their coherence, their similarity to the original, their human classification accuracy, and it does not even provide a list of example AEs.
> > >
> > > 2. Significantly more limited scope of the experimental setup
> > >
> > > Papernot et al. focus on a specific model (recurrent neural networks) and a specific task (sentiment analysis). Our work compares RNNs with word-level CNNs, character-level CNNs, and Naive Bayes. We also look at multiple tasks: sentiment analysis, fake news detection, spam classification. Our algorithm explicitly attempts to preserve of the meaning of the new sentences (and make the adversarial examples difficult to detect). We extensively evaluate our method on Mechanical Turk.
> > >
> > > Our algorithm greedily optimizes the objective (score of the wrong class); that of Papernot appears to optimize a linearization of that objective. We are happy to compare against their approach if you think it's important, but we don't see why this would be better than optimizing the actual objective function.
> > >
> > > Overall, we think that the paper by Papernot et al. does not prove that *imperceptible* adversarial examples can be constructed for text classification tasks. This is a crucial property of AEs. Our paper, on the other hand, demonstrates that it is possible.
> > >
> > >
> > >
> > > Finally, this paper was published at the "Military Communications Conference". We have never heard of this venue, and suspect that it might be closer to a peer-reviewed workshop in the machine learning community. The page count appears to be shorter, and the scope of the experiments seems to be considerably more narrow (e.g., human evaluation would be a must at ICML/NIPS/ICLR).
> > >
> > > As with the three papers below, we think it's a bit unfair to count previous recent publications on the Arxiv, conference workshops, or non-standard conference proceedings against the novelty of our paper. We believe that these papers should be cited and discussed as parallel work, but it is a bit harsh to claim that our work is incremental.

---

> ### Author Response · Authors · 2017-12-30
> **Have you had the chance to read our response?**
>
> Happy holidays! Since the end of the discussion period is approaching, we wanted to check if you had the chance to read our response. We would like very much to address your concerns and have a discussion about our paper before the January 5 deadline.
>
> Thank you!

---

### Official Review · AnonReviewer2 · 2017-12-01
**Ok paper, but needs some revision**

**Rating:** 4
**Confidence:** 4

**Review:**

The paper shows that neural networks are sensitive to adversarial perturbation for a set of NLP text classifications. They propose constructing (model-dependent) adversarial examples by optimizing a function J (that doesn't seem defined in the paper) subject to a constraint c(x, x') < \gamma (i.e. that the original input and adversarial input should be similar)

c is composed of two constraints:
1. || v - v' ||_2 < \gamma_1, where v and v' are bag of embeddings for each input
2. |log P(x') - log P(x)| < \gamma_2 where P is a language model

The authors then show that for 3 classification problems "Trec07p", "Yelp", and "News" and 4 models (Naive Bayes, LSTM, word CNNs, deep-char-CNNs) that the models that perform considerably worse on adversarial examples than on the test set.  Furthermore to test the validity of their adversarial examples, the authors show the following:
1. Humans achieve somewhat similar accuracy on the original adverarial examples (8 points higher on one dataset and 8 points lower on the other two)
2. Humans rate the writing quality of both the original and adversarial examples to be similar
3. The adversarial examples only somewhat transfer across models

My main questions/complaints/suggestions for the paper are:

-Novelty/Methodology. The paper has mediocre novelty given other similar papers recently.

On question I have is about whether the generated examples are actually close to the original examples. The authors do show some examples that do look good, but do not provide any systematic study (e.g. via human annotation)

 This is a key challenge in NLP (as opposed to vision where the inputs are continuous so it is easy to perturb them and be reasonably sure that the image hasn't changed much). In NLP however, the words are discrete, and the authors measure the difference between an original example and the adversary only in continuous space which may not actually be a good measure of how different they are.

They do have some constraint that the fraction of changed words cannot differ by more than delta, but delta = 0.5 in the experiments, which is really large! (i.e. 50% of the words could be different according to Algorithm 1)

-Writing: the function J is never mathematically defined, neither is the function c (except that it is known to be composed of the semantic/syntactic similarity constraints).

The authors talk about "syntactic" similarity but then propose a language model constraint. I think is a better word is "fluency" constraint.

The results in Table 3 and Table 6 seem different, shouldn't the diagonal of Table 6 line up with the results in Table 3?

-Experimental methodology (more of a question since authors are unclear): The authors write that "all adversarial examples are generated and evaluated on the test set".

There are many hyperparameters in the proposed authors' approach, are these also tuned on the test set? That is unfair to the base classifier. The adversarial model should be tuned on the validation set, and then the same model should be used to generate test set examples. (The authors can even show the validation adversarial accuracy to show how/if it deviates from the test accuracy)

-Lack of related work in NLP (see the anonymous comment for some examples). Even the related work in NLP that is cited e.g. Jia and Liang 2017 is obfuscated in the last page. The authors' introduction only refers to related works in vision/speech and ignores related NLP work.

Furthermore, adversarial perturbation is related to domain transfer  (since both involve shifts between the training and test distribution) and it is well known for instance that models that are trained on Wall Street Journal perform poorly on other domains.  See SJ Pan and Q Yang, A Survey on transfer learning, 2010, for some example references.

---

> ### Author Response · Authors · 2017-12-16
> **Addressing the novelty, methodology, and other concerns**
>
> Thank you for your feedback! We identified several concerns in your review.
>
> 1. Our work is not sufficiently novel.
>
> First, we believe that your claim that our paper has "mediocre novelty" is quite harsh. Especially given that your review does not include any references to related papers.
>
> Our paper explores adversarial examples for natural language classification. If you think this has been done before, could you please provide references? We will be more than happy to compare.
>
> Earlier, an anonymous commenter mentioned 3 references; we discuss these below.
>
> 2. We are not evaluating similarity between the adversarial examples and the originals.
>
> First, saying that we "don't provide any systematic study via human annotation" is incorrect: we measured both human accuracy and readability on every model/dataset combination (dozens of experiments in total).
>
> Next, we want to point out that the topic of similarity is more nuanced than it seems. Most often, the algorithm changes irrelevant parts of the input, e.g.:
>
> On Wednesday, Obama raised taxes (fake) -> On Tuesday, Obama raised taxes. (real)
> We ordered pasta and it was the worst we ever had (neg) -> We ordered chicken and it was the worst we ever had (pos)
>
> These are still valid similarity-based adversarial examples: i.e., we fool the fake news detection system and succeed in spreading the false news that Obama is raising taxes. What is most important is that humans and machines consistently classify our examples into opposite classes and the examples sound natural to humans.
>
> However, we understand the validity of your concern and we thank you for suggesting this experiment. To address your concern as much as possible, we performed the experiment in question.
>
> We quantified the similarity of adversarial examples via Mechanical Turk. We asked Turkers to rate the similarity of the adversarial examples to the originals on a scale of 1-5, with 1 being completely unrelated, and 5 being identical. Here are the results we compiled so far:
>
> Domain Score Number
> News 1 56
> News 2 49
> News 3 138
> News 4 141
> News 5 116
>
> Yelp 1 53
> Yelp 2 40
> Yelp 3 121
> Yelp 4 180
> Yelp 5 106
>
> Overall, we see that the majority of adversarial examples are similar to the originals.
>
> 3. We measure the difference of adversarial examples only in continuous space.
>
> Again, this is incorrect. We optimize a continuous objective; however we measure and report only metrics that are derived from human experiments (accuracy and readability)
>
> Although we set the maximum fraction of replaced words to 50%, we very rarely reach that number (see examples in the paper). This is just an early stopping criterion. Similarity is enforced via Equations 4 and 5, and the constants there are indeed tight. This can be seen by looking at the similarity of our examples to the originals. We are happy to add an experiment where we vary the threshold, if you think this is important.
>
> 4. Other technical issues
>
> The objective J is the score of the target (adversarial) class, and we define it right below  Equation 6. Sorry if this wasn't clear, we will make it more obvious.
>
> The function c is defined right below Equation 3, and is simply a vector of constraints. In our algorithm, we instantiate c with two constraints: a syntactical and a semantic one.
>
> We are going to think of a better name for the syntactic constraint (e.g., fluency as you suggested).
>
> We did not tune any hyper-parameters on the test set (we're not sure what might lead to think that). We chose hyper parameters on the training set (validation would have been slightly cleaner). We did not touch the set test, except for generating the final adversarial examples.
>
> We are certainly not “obfuscating” the work of Jia and Liang. We spend a whole paragraph comparing our work to theirs in Section 3.1. In brief, they create AEs by adding irrelevant sentences; we create AEs by changing some words to synonyms. We will extend the existing discussion if you think it’s necessary.

---

> ### Author Response · Authors · 2017-12-30
> **Have you had the chance to read our response?**
>
> Happy holidays! Since the end of the discussion period is approaching, we wanted to check if you had the chance to read our response. We would like very much to address your concerns and have a discussion about our paper before the January 5 deadline.
>
> Thank you!

---

### Public Comment · (anonymous) · 2017-11-05
**Lack of Related Works on NLP**

Interesting paper; however, it fails to cite related NLP papers. There is a vast amount of research on related topics, such as evading spam filters. Aside from that, adversarial examples for language has been also studied before. The followings are some related papers:

http://www.aclweb.org/anthology/W16-5603
https://arxiv.org/pdf/1702.08138.pdf
https://arxiv.org/pdf/1707.02812.pdf

---

> ### Author Response · Authors · 2017-12-16
> **Discussion of related work**
>
> Thank you for pointing us to these papers. We were not aware of them, and we agree that they need to be cited.
>
> However, these papers have not been published (they're workshop or arxiv papers) and they are very recent. They consider different and more limited settings than our work.
>
> We do not think it is fair to count this preliminary work against the novelty of our paper. In fact, these papers are most likely under peer review right now. We are going to cite and discuss the in the paper. We provide an outline of our discussion below.
>
> Hosseini et al.
> -------------------
>
> The main differences with this paper are:
> - They look at a very specific system and task: the Google Toxic Detection System
> - They obfuscate toxic words by mis-spelling them, e.g. stupid -> stu.pid
> - Their method tries random mis-spellings until the detection system is fooled.
>
> It is not clear what is the relevance of such specialized perturbations to the more general problem of adversarial examples, especially ones that are well-formed like in our paper. Our paper is substantially more general.
>
> Reddy et al.
> ----------------
>
> Here, the differences are:
> - They consider a very specific problem: fooling gender detection systems
> - They consider very specific classifiers: linear classifiers over bag-of-words feature counts
> - This enables them to substitute words that occur more frequently in one class relative to another.
>
> Like us, they also use word vectors to measure similarity between words. Their optimization algorithm is somewhat similar, but involves additional hand-crafted NLP rules.
>
> Overall, this paper considers a more limited setting than ours in terms of dataset and model. Again, it is recent and has not been published (it was presented at a workshop).
>
>
> Samanta et al.
> --------------------
>
> This paper is most similar to ours, and was released on ArXiv this summer while we were working on ours.
>
> It is the only one of the three to look at deep models (the same CNN architecture as we do). They also look at the same dataset.
>
> The differences are:
> - They use a more specialized algorithm with more specialized hand-crafted rules aimed at specific parts-of-speech.
> - They look at one setting (sentiment classification), while we look at two more (fake news and spam).
> - They look at CNNs, while we also look at LSTMs, character-level CNNs, and deep networks.
>
> Again, this should be considered as parallel work, but I don't think it is fair to count it against the novelty of our paper.

---

### Public Comment · ~Ryan_Knight1 · 2017-12-01
**Reproducibility Challenge**

As part of the reproducibility challenge, our team of students would like to attempt to reproduce the results of your paper.
If possible, it would be incredibly helpful if you could provide parts of the code used in your creation of the adversarial examples.

Also could you confirm if these are the datasets used for the paper:
Trec07p: https://plg.uwaterloo.ca/~gvcormac/treccorpus07/
Yelp: https://www.yelp.com/dataset
News: https://github.com/GeorgeMcIntire/fake_real_news_dataset/

And if this is the pre-trained word vector model that was used for word replacement in the LSTM method:
wor2vec: https://code.google.com/archive/p/word2vec/
while this pre-trained model was used for the other 3 methods:
GloVe: https://nlp.stanford.edu/projects/glove/

Thank you

---

### Public Comment · ~Matthew_Chan1 · 2017-12-16
**Reproducibility Challenge**

In the scope of the Reproducibility Challenge, the following is an executive summary of the findings on the paper "Adversarial Examples for Natural Language Classification Problems" currently under review for the ICLR 2018 Conference.

The paper provides the vast majority of the information required to reproduce the experiment. However, this is at a great expense in terms of the required time, effort and computational power. The lack of source code, hyperparameter specification and clean datasets are the driving factors behind the increased reproduction difficulty.

The models described in the paper were implemented using Keras with a Tensorflow backend and were trained on Google Cloud Computer Instances with 6 vCPUs, 24GB of RAM and either a NVidia K80 or a NVidia P100.

The datasets used in this experiment were publicly available in their raw form. Virtually no preprocessing was necessary to make the Yelp Polarity Dataset usable and a minimal amount was required for The News dataset. In contrast, cleaning the spam dataset proved to be challenging and it was ultimately abandoned in the final testing due to technical difficulties dealing with the encoding issues. It is possible that the dataset was exposed to antivirus software that may have tampered with the contents of the dataset and rendered them unusable. Although the procedure describing the data splitting and the preprocessing was clear, reproducing the exact same split in this experiment was impossible without a fixed random seed or the pre-split datasets (as was the case for the Yelp Polarity Dataset).

Without the source code and a full specification of the hyperparameters used to train the classifiers. It was challenging to even obtain optimal results on the clean data. This was likely caused by the time constraints of the reproducibility challenge which limited the quality of our hyperparameter search to achieve reasonable performance of the models. Our highest performing models for the Yelp Dataset achieve 86.79%, 79.64%, 70.21% and 56.12% for the Naive Bayes, the LSTM, the WCNN and VDCNN respectively. On the News dataset, the results were more comparable to the cited results, with each classifier achieving 87.68%, 91.24%, 85.17% and 51.10%. Although the hyper parameter tuning may have played a part in the relatively poor performance of the models on the Yelp Dataset, training the models for longer on the larger Yelp Dataset may have yielded better results. The reimplementation and training required a significant amount of time and effort.

The greedy algorithm's pseudocode clearly conveyed the protocol that was to be followed to emulate the specified behaviour. However, when implementing the algorithm, there were a few instances where assumptions had to be made. Notably, the optimization objective J(x') was not clearly specified in the paper. Although, the line "J(x') measures the extent to which x' is adversarial and may be a function of a target class y' != y, e.g. J(x') = f(x')" suggests that the optimization objective is the softmax value of the classifiers. Furthermore, it was assumed, with a relatively high degree of certainty, that the constraint function c(x, c')was merely a formalism to indicate that breaching any of the constraints should lead to a rejection of the candidate example. Optimizing the adversarial against each classifier led to poor results as observed in the paper. However, the human evaluation of the generated responses was not reproduced given the limited available resources.

Although the paper thoroughly specifies the architecture of the models and the details of the algorithm, reproducing the results of this paper would be greatly facilitated by providing source code along with the hyperparameter values of the classifiers. Furthermore, providing clean, split datasets reduces the effort required to preprocess the data, while ensuring that the reproducibility of the results will be more consistent. In conclusion, with more time, the results that were generated would have likely been more in line with the paper's published results but the time constraints imposed by the challenge were insufficient to do a more thorough hyperparameter search. Finally, the lack of resources which limited the ability to evaluate the generated adversarial examples by humans further reduced the credibility of this reproducibility experiment, seeing as the algorithm could be generating novel examples rather than true adversarial ones to fool the classifier.

---

### Public Comment · ~Li_X1 · 2017-12-16
**ICLR Reproducibility Challenge**

We are a group of students from McGill University who are reproducing the findings of your paper.

We implemented the Naïve Bayes Model and Word-Level CNN on the Real-Fake News dataset.
We generated adversarial examples following the individual word replacement model specified within the paper. However, we were not able to verify the hyperparameters used for lambda-1 (semantic similarity) and lambda-2 (syntactic similarity). However, we recognized that constraining for those factors were important, and so we elected to use WordNet’s synsets to find our replacement words.

We did two baseline models: Multinomial Naïve Bayes with sklearn and Naïve Bayes with nltk. Our testing accuracy: 88.8% (sklearn), 87.4% (nltk). In training, we shuffled the data and held aside 10% for testing. In testing, our accuracy was 69.2% and 65.4%, respectively. The drop in accuracy is significantly less than that of the authors.

We also implemented Word-Level CNN. The Kim paper referenced in by the authors had several adaptations. We checked all methods and picked the most robust one. The basic neural network is composed of a single embedding layer, a temporal convolution layer followed by max-pooling, and a fully connected layer for classification. The convolutional layer by max-pooling is composed of filters of three different sizes, where the filter windows are composed of 128 filter maps each. Filter sizes were chosen as [3, 3, 3], and [3, 4, 5] as reported by Kim. Regularization is done by performing Dropout on the before last layer of the network. The dropout rate is fixed at 0.50 is used with the Adam optimizer in tensorflow. Finally, training is done using stochastic gradient descent over shuffled mini-batches of size 64. Unfortunately, our computing resources did not allow us to use the Real-Fake News Dataset. Instead, we used a smaller dataset for a sentiment analysis task: Positive/Negative Movie Reviews. In training, we shuffled the data and held aside 10% for testing. Our training accuracy was 97.3%, and our testing accuracy was 72.7%.

From our reproduction, it’s evident that the execution of adversarial example generation highly influences the classifier’s ability to remain accurate. We followed the spirit of the authors’ adversarial example generation algorithm, but used a different implementation with WordNet (instead of word2vec or GloVe) to preserve semantic similarity.

---

> ### Author Response · Authors · 2017-12-30
> **Inaccuracies in how the paper was reproduced**
>
> Thank you for your interest in our paper and for your effort to reproduce some of our results.
>
> First, we want to point out that some of the choices you made when reproducing our paper are not quite accurate. Our paper uses the post-processed word vectors by Mrksic et al., which are crucial to replace words by their synonyms only. We did not make use of WordNet. Also, it is surprising that you were not able to match our accuracy on clean data, since these models have been shown to achieve similar accuracies in their original papers. There are also several more inaccuracies.
>
> We are happy to provide our source code as a file on an anonymous server. We are going to release this code with the camera-ready version of our paper, after we turn it into a form that is easy to read and execute.
>
> More generally, we think that reproducing published results is very important. However, some aspects of our method have not been exactly reproduced, which yields slightly different numbers. Unfortunately, this casts some doubt on the correctness of our paper (while it's under review), and we cannot release our source code to confirm our reported results (e.g., because of anonymity issues). Therefore, we think it would be best to validate reproducibility after the paper is published.

---

### Decision · Program_Chairs · 2018-01-29
**ICLR 2018 Conference Acceptance Decision**

**Decision:**

Reject

**Comment:**

This paper presents a way to generate adversarial examples for text classification.   The method is simple -- finding semantically similar words and replacing them in sentences with high language model score.  The committee identifies weaknesses in this paper that resonate with the reviews below -- reviewer 1 suggests that the authors should closely compare with the work of Papernot et al, and the response to that suggestion is not satisfactory.  Addressing such concerns would make the paper stronger for a future venue.